# A Credit Assignment Compiler for Joint Prediction

**Kai-Wei Chang**
University of Virginia
kw@kwchang.net

**He He**
University of Maryland
hhe@cs.umd.edu

**Hal Daumé III**
University of Maryland
me@hal3.name

**John Langford**
Microsoft Research
jcl@microsoft.com

**Stephane Ross**
Google
stephaneross@google.com

## Abstract

Many machine learning applications involve jointly predicting multiple mutually dependent output variables. Learning to search is a family of methods where the complex decision problem is cast into a sequence of decisions via a search space. Although these methods have shown promise both in theory and in practice, implementing them has been burdensomely awkard. In this paper, we show the search space can be defined by an arbitrary imperative program, turning learning to search into a credit assignment compiler. Altogether with the algorithmic improvements for the compiler, we radically reduce the complexity of programming and the running time. We demonstrate the feasibility of our approach on multiple joint prediction tasks. In all cases, we obtain accuracies as high as alternative approaches, at drastically reduced execution and programming time.

## 1 Introduction

Many applications require a predictor to make coherent decisions. As an example, consider recognizing a handwritten word where each character might be recognized in turn to understand the word. Here, it is commonly observed that exposing information from related predictions (i.e. adjacent letters) aids individual predictions. Furthermore, optimizing a joint loss function can improve the gracefulness of error recovery. Despite these advantages, it is empirically common to build independent predictors, in settings where joint prediction naturally applies, because they are simpler to implement and faster to run. Can we make joint prediction algorithms as easy and fast to program and compute while maintaining their theoretical benefits?

Methods making a sequence of sub-decisions have been proposed for handling complex joint predictions in a variety of applications, including sequence tagging [30], dependency parsing (known as transition-based method) [35], machine translation [18], and co-reference resolution [44]. Recently, general search-based joint prediction approaches (e.g., [10, 12, 14, 22, 41]) have been investigated. The key issue of these search-based approaches is credit assignment: when something goes wrong do you blame the first, second, or third prediction? Existing methods often take two strategies:

- The system ignores the possibility that a previous prediction may have been wrong, different costs have different errors, or the difference between train-time and test-time prediction.

- The system uses handcrafted credit assignment heuristics to cope with errors that the underlying algorithm makes and the long-term outcomes of decisions.

Both approaches may lead to statistical inconsistency: when features are not rich enough for perfect prediction, the machine learning may converge sub-optimally.

---
**Algorithm 1** MYRUN(*X*) % for sequence tagging, X: input sequence, Y: output
---
    A sample user-defined function, where PREDICT and LOSS are library functions (see text). The credit assignment compiler translates the code and data into model updates. More examples are in appendices.

1:   $Y \leftarrow$ []
2:   **for** $t$ = $1$ to LEN($X$) **do**
3:      $ref \leftarrow X[t]$.true_label
4:      $Y[t] \leftarrow$ PREDICT(x=*examples*[t], y=*ref*, tag=*t*, condition=[1:*t*-1])
5:   LOSS(number of $Y[t] \neq X[t]$.true_label)
6:   **return** $Y$
---

In contrast, learning to search approaches [5, 11, 40] automatically handle the credit assignment problem by decomposing the production of the joint output in terms of an explicit search space (states, actions, etc.); and learning a control policy that takes actions in this search space. These have formal correctness guarantees which differ qualitatively from models such as Conditional Random Fields [28] and structured SVMs [46, 47]. Despite the good properties, none of these methods have been widely adopted because the specification of a search space as a finite state machine is awkward and naive implementations do not fully demonstrate the ability of these methods.

In this paper, we cast learning to search into a credit assignment compiler with a new programming abstraction for representing a search space. Together with several algorithmic improvements, this radically reduces both the complexity of programming[1] and the running time. The programming interface has the following advantages:

- The same decoding function (see Alg. 1 for example) is used for training and prediction so a developer need only code desired test time behavior and gets training "for free". This simple implementation prevents common train/test asynchrony bugs.
- The compiler automatically ensures the model learns to avoid compounding errors and makes a sequence of coherent decisions.
- The library functions are in a reduction stack so as base classifiers and learning to search approaches improve, so does joint prediction performance.

We implement the credit assignment compiler in Vowpal-Wabbit (http://hunch.net/~vw/), a fast online learning library, and show that the credit assignment compiler achieves outstanding empirical performance both in accuracy and in speed for several application tasks. This provides strong simple baselines for future research and demonstrates the compiler approach to solving complex prediction problems may be of broad interest. Details experimental settings are in appendices.

## 2   Programmable Learning to Search

We first describe the proposed programmable joint prediction paradigm. Algorithm 1 shows sample code for a part of speech tagger (or generic sequence labeler) under Hamming loss. The algorithm takes as input a sequence of examples (e.g., words), and predicts the meaning of each element in turn. The $i^{th}$ prediction depends on previous predictions.[2] It uses two underlying library functions, PREDICT(...) and LOSS(...). The function PREDICT(...) returns individual predictions based on $x$ while LOSS(...) allows the declaration of an *arbitrary* loss for the point set of predictions. The LOSS(...) function and the reference $y$ inputted to PREDICT(...) are only used in the training phase and it has no effect in the test phase. Surprisingly, this single library interface is sufficient for both testing *and training*, when augmented to include label "advice" from a training set as a reference decision (by the parameter $y$). This means that a developer only has to specify the desired *test time behavior* and gets training with minor additional decoration. The underlying system works as a credit assignment compiler to translate the user-specified decoding function and labeled data into updates of the learning model.

How can you learn a good PREDICT function given just an imperative program like Algorithm 1? In the following, we show that it is essential to run the MYRUN(...) function (e.g., Algorithm 1) many times, "trying out" different versions of PREDICT(...) to learn one that yields low LOSS(...). We begin with formal definitions of joint prediction and a search space.

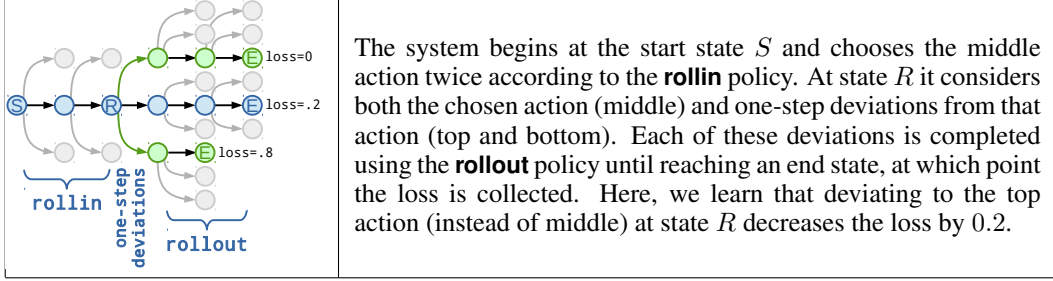

The system begins at the start state $S$ and chooses the middle action twice according to the **rollin** policy. At state $R$ it considers both the chosen action (middle) and one-step deviations from that action (top and bottom). Each of these deviations is completed using the **rollout** policy until reaching an end state, at which point the loss is collected. Here, we learn that deviating to the top action (instead of middle) at state $R$ decreases the loss by $0.2$.

Figure 1: A search space implicitly defined by an imperative program.

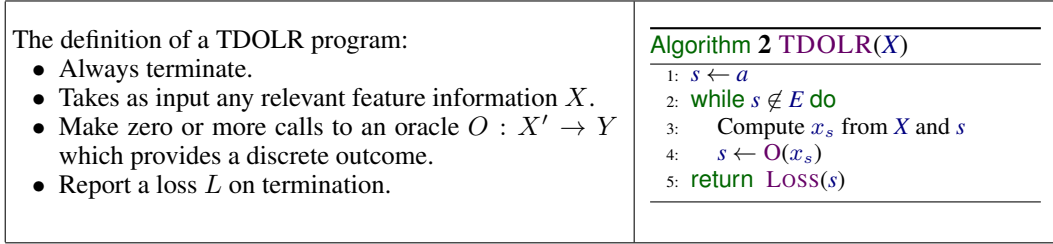

Figure 2: Left: the definition; right: A TDOLR program simulates the search space.

**Joint Prediction.** Joint prediction aims to induce a function $f$ such that for any $X \in \mathcal{X}$ (the input space), $f$ produces an output $f(X) = Y \in \mathcal{Y}(X)$ in a (possibly input-dependent) space $\mathcal{Y}(X)$. The output $Y$ often can be decomposed into smaller pieces (e.g., $y_1, y_2, \ldots$), which are tied together by features, by a loss function and/or by statistical dependence. There is a task-specific loss function $\ell : \mathcal{Y} \times \mathcal{Y} \to \mathbb{R}^{\geq 0}$, where $\ell(Y^*, \hat{Y})$ tells us how bad it is to predict $\hat{Y}$ when the true is $Y^*$.

**Search Space.** In our framework, the joint variable $\hat{Y}$ is produced incrementally by traversing a search space, which is defined by states $s \in S$ and a mapping $A : S \to 2^S$ defining the set of valid next states.[3] One of the states is a unique start state $S$ while some of the others are end states $e \in E$. Each end state corresponds to some output variable $Y_e$. The goal of learning is finding a function $f : X_s \to S$ that uses the features of an input state ($x_s$) to choose the next state so as to minimize the loss $\ell(Y^*, Y_e)$ on a holdout test set.[4] Follow reinforcement learning terminology, we call the function a policy and call the learned function $f$ a learned policy $\pi_f$.

**Turning Search Space into an Imperative Program** Surprisingly, search space can be represented by a class of imperative program, called Terminal Discrete Oracle Loss Reporting (TDOLR) programs. The formal definition of TDOLR is listed in Figure 2. Without loss of generality, we assume the number of choices is fixed in a search space, and the following theorem holds:

**Theorem 1.** *For every TDOLR program, there exist an equivalent search space and for every search space, there exists an equivalent TDOLR program.*

*Proof.* A search space is defined by $(A, E, S, l)$. We show there is a TDOLR program which can simulate the search space in algorithm 2. This algorithm does a straightforward execution of the search space, followed by reporting of the loss on termination. This completes the second claim. For the first claim, we need to define, $(A, E, S, l)$ given a TDOLR program such that the search space can simulate the TDOLR program. At any point in the execution of TDOLR, we define an equivalent state $s = (O(X_1), ..., O(X_n))$ where $n$ is the number of calls to the oracle. We define $a$ as the sequence of zero length, and we define $E$ as the set of states after which TDOLR terminates.

**Algorithm 3** LEARN($X$, F)

1:   *T, ex, cache* ← *0*, [], []
2:   define PREDICT($x$, $y$) := { $T$++ ; $ex[T$-1$]$ ← $x$; $cache[T$-1$]$ ← F($x$, $y$, rollin) ; return  $cache[T$-1$]$  }
3:   define LOSS($l$) := no-op
4:   MYRUN(X) % MYRUN(X) is a user-defined TDOLR program (e.g., Algorithm 1).
5:   for $t_0 = 1$ to $T$ do
6:     *losses, t* ← $\langle 0, 0, \ldots, 0 \rangle$, *0*
7:     for $a_0 = 1$ to A($ex[t_0]$) do
8:       Define PREDICT($x$, $y$) := { $t$++ ; return $\begin{cases} cache[t\text{-}1] & \text{if } t < t_0 \\ a_0 & \text{if } t = t_0 \\ \text{F}(x,y,\text{rollout}) & \text{if } t > t_0 \end{cases}$ }
9:       Define LOSS(*val*) := { *losses*[$a_0$] += *val* }
10:      MYRUN(X)
11:     Online update with cost-sensitive example ($ex[t_0]$, *losses*)

---

For each $s \in E$ we define $l(s)$ as the loss reported on termination. This search space manifestly outputs the same loss as the TDOLR program.     □

The practical implication of this theorem is that instead of specifying search spaces, we can specify a TDOLR program (e.g., Algorithm 1), reducing the programming complexity of joint prediction.

## 3   Credit Assignment Compiler for Training Joint Predictor

Now, we show how a credit assignment compiler turns a TDOLR program and training data into model updates. In the training phase, the supervised signals are used in two places: 1) to define the loss function, and 2) to construct a reference policy $\pi^*$. The reference policy returns at any prediction point a "suggestion" as to a good next state.[5] The general strategy is, for some number of epochs, and for each example $(X, Y)$ in the training data, to do the following:

1. Execute MYRUN(...) on $X$ with a **rollin policy** to obtain a trajectory of actions $\vec{a}$ and loss $\ell_0$
2. Many times:
   - (a) For some (or for all) time step $t \leq |\vec{a}|$
   - (b) For some (or for all) alternative action $a'_t \neq a_t$ ($a_t$ is the action taken by $\vec{a}$ in time step $t$)
   - (c) Execute MYRUN(...) on $X$, with PREDICT returning $a_{1:t-1}$ initially, then $a'_t$, then acting according to a **rollout policy** to obtain a new loss $\ell_{t,a'_t}$
   - (d) Compare the overall losses $\ell_{t,a_t}$ and $\ell_{t,a'_t}$ to construct a classification/regression example that demonstrates how much better or worse $a'_t$ is than $a_t$ in this context.
3. Update the learned policy

The rollin and rollout policies can be the reference $\pi^*$, the current classifier $\pi_f$ or a mixture between them. By varying them and the manner in which classification/regression examples are created, this general framework can mimic algorithms like SEARN [11], DAGGER [41], AGGREVATE [40], and LOLS [5].[6]

The full learning algorithm (for a single joint input $X$) is depicted in Algorithm 3.[7] In lines 1–4, a *rollin* pass of MYRUN is executed. MYRUN can generally be any TDOLR program as discussed (e.g., Alg. 1). In this pass, predictions are made according to the current policy, F, flagged as rollin (this is to enable support of arbitrary rollin and rollout policies). Furthermore, the examples (feature vectors) encountered during prediction are stored in *ex*, indexed by their position in the sequence ($T$), and the rollin predictions are cached in the variable *cache* (see Sec. 4).

The algorithm then initiates one-step deviations from this rollin trajectory. For every time step, (line 5), we generate a *single* cost-sensitive classification example; its features are $ex[t_0]$, and there

are $A(ex[t_0])$ possible labels (=actions). For each action (line 7), we compute the *cost* of that action by executing MYRUN again (line 10) with a "tweaked" PREDICT which returns the cached predictions at steps before $t_0$, returns the perturbed action $a_0$ at $t_0$, and at future timesteps calls F for rollouts. The LOSS function accumulates the loss for the query action. Finally, a cost-sensitive classification example is generated (line 11) and fed into an online learning algorithm.

## 4   Optimizing the Credit Assignment Compiler

We present two algorithmic improvements which make training orders of magnitude faster.

**Optimization 1: Memoization**   The primary computational cost of Alg. 3 is making predictions: namely, calling the underlying classifier in Step 10. In order to avoid redundant predictions, we cache previous predictions. The challenge is understanding how to know when two predictions are going to be identical, faster than actually computing the prediction. To accomplish this, the user may decorate calls to the PREDICT function with *tags*. For a graphical model, a tag is effectively the "name" of a particular variable in the graphical model. For a sequence labeling problem, the tag for a given position might just be its index. When calling PREDICT, the user specifies both the tag of the current prediction and the tag of *all previous predictions* on which the current prediction depends. The user is guaranteeing that *if* the predictions for all the tags in the dependent variables are the same, *then* the prediction for the current example are the same.

Under this assumption, we store a cache that maps triples of ⟨tag, condition tags, condition predictions⟩ to ⟨current prediction⟩. The added overhead of maintaining this data structure is tiny in comparison to making repeated predictions on the same features. In line 11 the learned policy changes making correctness subtle. For data mixing algorithms (like DAgger), this potentially changes $F_i$ implying the memoized predictions may no longer be up-to-date. Thus this optimization is okay *if* the policy does not change much. We evaluate this empirically in Section 5.3.

**Optimization 2: Forced Path Collapse**   The second optimization we can use is a heuristic that only makes rollout predictions for a constant number of steps (e.g., 2 or 4). The intuition is that optimizing against a truly long term reward may be impossible if features are not available at the current time $t_0$ which enable the underlying learner to distinguish between the outcome of decisions far in the future. The optimization stops rollouts after some fixed number of rollout steps.

This intuitive reasoning is correct, except for accumulating LOSS(...). If LOSS(...) is only declared at the end of MYRUN, then we must execute $T-t_0$ time steps making (possibly memoized) predictions. However, for many problems, it is possible to declare loss *early* as with Hamming loss (= number of incorrect predictions). There is no need to wait until the end of the sequence to declare a per-sequence loss: one can declare it after every prediction, and have the total loss accumulate (hence the "+=" on line 9). We generalize this notion slightly to that of a history-independent loss:

**Definition 1** (History-independent loss). *A loss function is* history-independent at state $s_0$ *if, for any final state $e$ reachable from $s_0$, and for any sequence $s_0 s_1 s_2 \ldots s_i = e$: it holds that* $\text{LOSS}(e) = A(s_0) + B(s_1 s_2 \ldots s_i)$, *where $B$ does not depend on any state before $s_1$.*

For example, Hamming loss is history-independent: $A(s_0)$ corresponds to loss through $s_0$ and $B(s_1 \ldots s_i)$ is the loss after $s_0$.[8] When the loss function being optimized is history-independent, we allow LOSS(...) to be declared early for this optimization. In addition, for tasks like transition-base dependency parsing, although LOSS(...) is not decomposable over actions, expected cost per action can be directly computed based on gold labels [19] so the array *losses* can be directly specified.

**Speed Up**   We analyze the time complexity of the sequence tagging task. Suppose that the cost of calling the policy is $d$ and each state has $k$ actions.[9] Without any speed enhancements, each execution of MYRUN takes $\mathcal{O}(T)$ time, and we execute it $Tk + 1$ times, yielding an overall complexity of $\mathcal{O}(kT^2 d)$ per joint example. For comparison, structured SVMs or CRFs with first order Markov

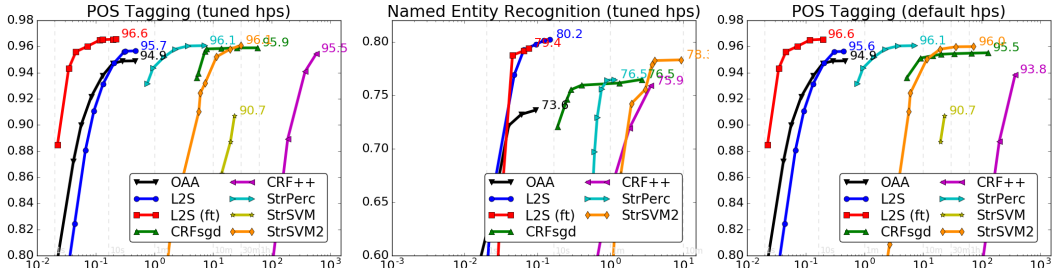

Figure 3: Training time (minutes) versus test accuracy for POS and NER. Different points correspond to different termination criteria for training. The rightmost figure use *default* hyperparameters and the two left figures use hyperparameters that were tuned (for accuracy) on the holdout data. Results of NER with default parameters are in the appendix. **X-axis is in log scale.**

dependencies run in $\mathcal{O}(k^2T)$ time. When both memoization and forced path collapse are in effect, the complexity of training drops to $\mathcal{O}(Tkd)$, similar to independent prediction. In particular, if the $i^{th}$ prediction only depends on the $i-1^{th}$ prediction, then at *most* $Tk$ unique predictions are made.[10]

# 5    System Performance

We present two sets of experiments. In the first set, we compare the credit assignment compiler with existing libraries on two sequence tagging problems: Part of Speech tagging (POS) on the Wall Street Journal portion of the Penn Treebank; and sequence *chunking* problem: named entity recognition (NER) based on standard Begin-In-Out encoding on the CoNLL 2003 dataset. In the second set of experiments, we demonstrate a simple dependency parser built by our approach achieves strong results when comparing with systems with similar complexity. The parser is evaluated on the standard WSJ (English, Stanford-style labels), CTB (Chinese) datasets and the CoNLL-X datasets for 10 other languages.[11] Our approach is implemented using the Vowpal Wabbit [29] toolkit on top of a cost-sensitive classifier [3] trained with online updates [15, 24, 42]. Details of dataset statistics, experimental settings, additional results on other applications, and pseudocode are in the appendix.

## 5.1    Sequence Tagging Tasks

We compare our system with freely available systems, including CRF++ [27], CRF SGD [4], Structured Perceptron [9], Structured SVM [23], Structured SVM (DEMI-DCD) [6], and an *unstructured* baseline (OAA) predicting each label independently, using one-against-all classification [3][12].

For each system, we consider two situations, either the **default hyperparameters** or the **tuned hyperparameters** that achieved the best performance on holdout data. We report both conditions to give a sense of how sensitive each approach is to the setting of hyperparameters (the amount of hyperparameter tuning directly affects effective training time). We use the built-in *feature template* of CRF++ to generate features and use them for other systems. The templates included neighboring words and, in the case of NER, neighboring POS tags. The CRF++ templates generate $630k$ unique features for the training data. *However,* because L2S is also able to generate features from its own templates, we also provide results for **L2S (ft)** in which it uses its own feature template generation.

**Training time.**    In Figure 3, we show trade-offs between training time (x-axis, log scaled) and prediction accuracy (y-axis) for the aforementioned six systems. For POS tagging, the independent classifier is the fastest (trains in less than one minute) but its performance peaks at $95\%$ accuracy. Three other approaches are in roughly the same time/accuracy trade-off: L2s, L2S (ft) and Structured Perceptron. CRF SGD takes about twice as long. DEMI-DCD (taking a half hour) and CRF++ (taking

| Parser | AR | BU | CH | CZ$^+$ | DA | DU$^+$ | JA$^+$ | PO$^+$ | SL$^+$ | SW | PTB | CTB |
|---|---|---|---|---|---|---|---|---|---|---|---|---|
| DYNA | 75.3 | 89.8 | 88.7 | 81.5 | 87.9 | 74.2 | **92.1** | 88.9 | 78.5 | 88.9 | 90.3 | 80.0 |
| SNN | 67.4* | 88.1 | 87.3 | 78.2 | 83.0 | 75.3 | 89.5 | 83.2* | 63.6* | 85.7 | 91.8$^\#$ | 83.9$^\#$ |
| L2S | **78.2** | **92.0** | **89.8** | **84.8** | **89.8** | **79.2** | 91.8 | **90.6** | **82.2** | **89.7** | **91.9** | **85.1** |
| BEST | 79.3 | 92.0 | 93.2 | 87.30 | 90.6 | 83.6 | 93.2 | 91.4 | 83.2 | 89.5 | 94.4$^\#$ | 87.2$^\#$ |

Table 1: UAS on PTB, CTB and CoNLL-X. Best: the best known result in CoNLL-X or the best published results (CTB, PTB) using arbitrary features and resources. See details and additional results in text and in the appendix.[15]

over five hours) are not competitive. Structured SVM runs out of memory before achieving competitive performance (likely due to too many constraints). For NER the story is a bit different. The independent classifiers are not competitive. Here, the two variants of L2S totally dominate. In this case, Structured Perceptron is no longer competitive and is essentially dominated by CRF SGD. The only system coming close to L2S's performance is DEMI-DCD, although it's performance flattens out after a few minutes.[13] The trends in the runs with default hyperparameters show similar behavior to those with tuned, though some of the competing approaches suffer significantly in prediction performance. Structured Perceptron has no hyperparameters.

**Test Time.** In addition to training time, one might care about test time behavior. On NER, prediction times where $5.3k$ tokens/second (DEMI-DCD and Structured Perceptron, $20k$ (CRF SGD and Structured SVM), $100k$ (CRF++), $220k$ (L2S (ft)), and $285k$ (L2S). Although CRF SGD and Structured Perceptron fared well in terms of training time, their test-time behavior is suboptimal. When the number of labels increases from 9 (NER) to 45 (POS) the relative advantage of L2S increases further. The speed of L2S is about halved while for others, it is cut down by as much as a factor of 8 due to the $O(k)$ vs $\mathcal{O}(k^2)$ dependence on the label set size.

## 5.2 Dependency Parsing

To demonstrate how the credit assignment compiler handles predictions with complex dependencies, we implement an arc-eager transition-based dependency parser [35]. At each state, it takes one of the four actions $\{Shift, Reduce, Left, Right\}$ based on a simple neural network with one hidden layer of size 5 and generates a dependency parse to a sentence in the end. The rollin policy is the current (learned) policy. The probability of executing the reference policy (dynamic oracle) [19] for rollout decreases over each round. We compare our model with two recent greedy transition-based parsers implemented by the original authors, the dynamic oracle parser (DYNA) [19] and the Stanford neural network parser (SNN) [8]. We also present the best results in CoNLL-X and the best-published results for CTB and PTB. The performances are evaluated by unlabeled attachment scores (UAS). Punctuation is excluded.

Table 1 shows the results. Our implementation with only ~300 lines of C++ code is competitive with DYNA and SNN, which are specifically designed for parsing. Remarkably, our system achieves strong performance on CoNLL-X without tuning any hyper-parameters, even beating heavily tuned systems participating in the challenge on one dataset. The best system to date on PTB [2] uses a global normalization, more complex neural network layers and k-best POS tags. Similarly, the best system for CTB [16] uses stack LSTM architectures tailored for dependency parsing.

## 5.3 Empirical evaluation of optimizations

In Section 3, we discussed two approaches for computational improvements. Memoization avoids re-predicting on the same input multiple times while path collapse stops rollouts at a particular

|          | NER | | POS | |
|----------|------|-------|-------|-------|
|          | LOLS | Searn | LOLS | Searn |
| **No Opts** | 96s | 123s | 3739s | 4255s |
| **Mem.** | 75s | 85s | 1142s | 1215s |
| **Col.@4+Mem.** | 71s | 75s | 1059s | 1104s |
| **Col.@2+Mem.** | 69s | 71s | 1038s | 1074s |

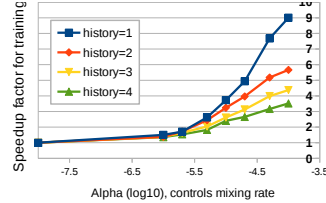

Figure 4: The table on the left shows the effect of Collapse (Col) and Memoization (Mem.). The figure on the right shows the speed-up obtained for different historical lengths and mixing rate of rollout policy. Large $\alpha$ corresponds to more prediction required when training the model.

point in time. The effect of the different optimizations depends greatly on the underlying learning algorithm. For example, DAgger does not do rollouts at all, so no efficiency is gained by either optimization.[16] The affected algorithms are LOLS (with mixed rollouts) and Searn.

Figure 4 shows the effect of these optimizations on the *best* NER and POS systems we trained without using external resources. In the left table, we can see that memoization alone reduces overall training runtime by about $25\%$ on NER and about $70\%$ on POS, essentially because the overhead for the classifier on POS tagging is so much higher (45 labels versus 9). When rollouts are terminated early, the speed increases are much more modest, essentially because memoization is already accounting for much of these gains. In all cases, the final performance of the predictors is within statistical significance of each other (p-value of 0.95, paired sign test), *except* for Collapse@2+Memoization on NER, where the performance decrease is only insignificant at the 0.90 level. The right figure demonstrates that when $\alpha$ increases, more prediction is required during the training time, and the speedup increases from a factor of 1 (no change) to a factor of as much as 9. However, as the history length increases, the speedup is more modest due to low cache hits.

## 6 Related Work

Several algorithms are similar to learning to search approaches, including the incremental structured perceptron [10, 22], HC-Search [13, 14], and others [12, 38, 45, 48, 49]. Some fit this framework.

Probabilistic programming [21] has been an active area of research. These approaches have a different goal: Providing a flexible framework for specifying graphical models and performing inference in those models. The credit assignment compiler instead allows a developer to learn to make coherent decisions for joint prediction ("learning to search"). We also differ by not designing a new programming language. Instead, we have a two-function library which makes adoption and integration into existing code bases much easier.

The closest work to ours is Factorie [31]. Factorie is essentially an embedded language for writing factor graphs compiled into Scala to run efficiently.[17] Similarly, Infer.NET [33], Markov Logic Networks (MNLs) [39], and Probabilistic Soft Logic (PSL) [25] concisely construct and use probabilistic graphical models. BLOG [32] falls in the same category, though with a very different focus. Similarly, Dyna [17] is a related declarative language for specifying probabilistic dynamic programs, and Saul [26] is a declarative language embedded in Scala that deals with joint prediction via integer linear programming. All of these examples have picked particular aspects of the probabilistic modeling framework to focus on. Beyond these examples, there are several approaches that essentially "reinvent" an existing programming language to support probabilistic reasoning at the first order level. IBAL [36] derives from O'Caml; Church [20] derives from LISP. IBAL uses a (highly optimized) form of variable elimination for inference that takes strong advantage of the structure of the program; Church uses MCMC techniques, coupled with a different type of structural reasoning to improve efficiency.

**Acknowledgements** Part of this work was carried out while Kai-Wei, Hal and Stephane were visiting Microsoft Research. Hal and He are also supported by NSF grant IIS-1320538. Any opinions, findings, conclusions, or recommendations expressed here are those of the authors and do not necessarily reflect the view of the sponsor. The authors thank anonymous reviewers for their comments.

## Footnotes

[1]With library supports, developing a new task often requires only a few lines of code.

[2]In this example, we use the library's support for generating implicit features based on previous predictions.

[3]Comprehensive strategies for defining search space have been discussed [14]. The theoretical properties do not depend on which search space definition is used.

[4]Note that we use $X$ and $Y$ to represent joint input and output and use $x$ and $y$ to represent input and output to function $f$ and PREDICT.

[5]Some papers assume the reference policy is optimal. An optimal policy always chooses the best next state assuming it gets to make all future decisions as well.

[6]E.g., rollin in LOLS is $\pi_f$ and rollout is a stochastic interpolation of $\pi_f$ and oracle $\pi^*$ constructed by $y$.

[7]This algorithm is awkward because standard computational systems have a single stack. We have elected to give MYRUN control of the stack to ease the implementation of joint prediction tasks. Consequently, the learning algorithm does not have access to the machine stack and must be implemented as a state machine.

[8] Any loss function that decomposes over the structure, as required by structured SVMs, is guaranteed to also be history-independent; the reverse is not true. Furthermore, when structured SVMs are run with a non-decomposable loss function, their runtime becomes exponential in $t$. When our approach is used with a loss function that's not history-independent, our runtime increases by a factor of $t$.

[9] Because the policy is a multiclass classifier, $d$ might hide a factor of $k$ or $\log k$.

[10]We use *tied randomness* [34] to ensure that for any time step, the same policy is called.

[11]PTB and CTB are prepared by following [8], and CoNLL-X is from the CoNLL shared task 06.

[12] Structured Perceptron and Structured SVM (DEMI-DCD) are implemented in Illioins-SL[7]. DEMI-DCD is a multi-core dual approach, while Structured SVM uses cutting-planes.

[13]We also tried giving CRF SGD the features computed by L2S (ft) on both POS and NER. On POS, its accuracy improved to 96.5 with essentially the same speed. On NER it's performance decreased.

[15](*) SNN makes assumptions about the structure of languages and hence obtains substantially worse performance on languages with multi-root trees. ($^+$) Languages contains more than 1% non-projective arcs, where a transition-based parser (e.g. L2S) likely underperforms graph-based parser (Best) due to the model assumptions. ($^\#$) Numbers reported in the published papers [8, 16, 2].

[16]Training speed is only degraded by about 0.5% with optimizations on, demonstrating negligible overhead.

[17]Factorie-based implementations of simple tasks are still less efficient than systems like CRF SGD.

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
