[Supplementary Material]

---

**Algorithm 4** SEQUENTIAL_DETECTION_RUN(*examples* as $X$, *false_negative_loss*)

---

1: Let *max_value* = 1
2: **for** $i = 1$ **to** LEN(*examples*) **do**
3:    *max_value* ← MAX(*max_value*, *examples*[*i*].*label*)
4: Let *max_prediction* = 1
5: **for** $i = 1$ **to** LEN(*examples*) **do**
6:    *max_prediction* ← MAX(*max_prediction*, PREDICT(*x*=*examples*[*i*], *y*=*examples*[*i*].*label*))  // maintain max
7: **if** *max_label* > *max_prediction* **then**
8:    LOSS(*false_negative_loss*)                                               // The loss is asymmetric
9: **else**
10:    **if** *max_label* < *max_prediction* **then**
11:       LOSS(1)
12:    **else**
13:       LOSS(0)
14: **if** *output*.*good* **then**
15:    *output* « *max_prediction*               // if we should generate output, append our prediction

---

**Algorithm 5** ENTITY_RELATION_RUN(*sent* as $X$)

---

1: *output* ← INITIALIZE_STRUCTURE()
2: $K$ ← NUMBER_OF_ENTITIES(*sent*)
3: **for** $n = 1$ **to** $K$ **do**
4:    *ref* ← *sent*.*entity_type*[*n*]
5:    *output*.*entity_type[n]* ← PREDICT(*x*=*sent*.entity[i], *y*=*ref*, tag=*n*)
6:    LOSS(*output*.entity_type[n] ≠ *sent*.entity_type[n].true_label)
7: **for** $n = 1$ **to** $K$-1 **do**
8:    **for** $m = n+1$ **to** $K$) **do**
9:       *ref* ← *sent*.relation_type[*n*,*m*].true_label
10:       *valid_relations* ← FIND_VALID_RELATIONS(*output*.entity_type[*n*], *output*.entity_type[*m*])
11:       *output*.*relation_type* ← PREDICT(*x*=*sent*.relation[n,m], *y*=*ref*, tag=*K*\*(*n*+1)+*m*, valid_labels=*valid_relations*, condition=[*n*,*m*])
12:       LOSS(*output*.relation_type[n,m] ≠ *sent*.entity_type[n,m].true_label )
13: **return** *output*

---

# Supplement: Credit Assignment Compiler for Joint Prediction

## A   Example TDOLR programs

In this section, a few other TDOLR programs which illustrate the ease and flexibility of programming.

Algorithm 4 is for a sequential *detection* task where the goal is to detect whether or not a sequence contains some rare element. This illustrates outputs of lengths other than the number of examples, explicit loss functions.

Algorithm 5 provides an implementation for jointly assigning types to name entities in a sentence and recognizing relations between them [43]. Besides features used for predicting entity and relation types. We also consider constraints that ensure the entity-type assignments and relation-type assignments are compatible with each other. For example, the first argument of the work_for relation need to be tagged as person, and the second argument has to be an organization.

Finally, in Algorithm 6, we show an implementation of a shift-reduce dependency parser for natural language. We discuss each subcomponent below. The detailed introduction to dependency parsing is provided in the next section.

- GETVALIDACTION returns valid actions that can be taken by the current configuration.

- GETFEAT extracts features based on the current configuration. A detailed list of our features is in the supplementary material.

---

Algorithm 6 RUNPARSER(*sentence* as $X$)

1: *stack $S$* ← {**Root**}
2: *buffer $B$* ← [words in sentence]
3: *arcs $A$* ← ∅
4: **while** $B \neq \emptyset$ or $|S| > 1$ **do**
5:     *ValidActs* ← GETVALIDACTIONS($S, B$)
6:     *features* ← GETFEAT($S, B, A$)
7:     *ref* ← GETGOLDACTION($S, B$)
8:     *action* ← PREDICT($x$=features, $y$=ref, ValidActs)
9:     $S, B, A$ ← TRANS($S, B, A$, action)
10: LOSS($A[w] \neq A^*[w], \forall w \in$ sentence)
11: **return** output

---

| Action | Configuration | | |
|---|---|---|---|
| | Stack Buffer | | Arcs |
| | [**Root**] [Flying planes can be dangerous] | | {} |
| SHIFT | [**Root** Flying] [planes can be dangerous] | | {} |
| REDUCE-LEFT | [**Root**] [planes can be dangerous] | | {(planes, Flying)} |
| SHIFT | [**Root** planes] [can be dangerous] | | {(planes, Flying)} |
| REDUCE-LEFT | [**Root**] [can be dangerous] | | {(planes, Flying), (can, planes)} |
| SHIFT | [**Root** can] [be dangerous] | | {(planes, Flying), (can, planes)} |
| SHIFT | [**Root** can be] [dangerous] | | {(planes, Flying), (can, planes)} |
| SHIFT | [**Root** can be dangerous] [] | | {(planes, Flying), (can, planes)} |
| REDUCE-RIGHT | [**Root** can be] [] | | {(planes, Flying), (can, planes), (be, dangerous)} |
| REDUCE-RIGHT | [**Root** can] [] | | {(planes, Flying), (can, planes), (be, dangerous), (can, be)} |
| REDUCE-RIGHT | [**Root**] [] | | {(planes, Flying), (can, planes), (be, dangerous), (can, be), (**Root**, can)} |

Parse tree derived by the above parser          Gold parse tree

Figure 5: An illustrative example of an arc-hybrid transition parser. The above table show the actions taken and the intermediate configurations generated by a parser. The parse tree derived by the parser is in the bottom left, and the gold parse tree is the bottom right. The distance between these two trees is 2.

- GETGOLDACTION implements the dynamic oracle described in [19]. The dynamic oracle returns the optimal action in any state that leads to a reachable end state with the minimal loss.

- PREDICT is a library call implemented in the L2S system. Given training samples, L2S can learn the policy automatically. In the test phase, it returns a predicted action leading to an end state with small structured loss.

- TRANS implements the hybrid-arc transition system described above.

- LOSS returns the number of words whose parents are wrong. It has no effect in the test phase.

We show that this parser achieves strong results across ten languages from the CoNLL-X challenge and performs well on two standard evaluation data sets, and requires only about 300 lines of readable C++ code.

## B  Dependency Parsing

In the following, we provide a brief overview of transition-based dependency parsing. A transition-based dependency parser takes a sequence of actions and parses a sentence from left to right by maintaining a *stack $S$*, a *buffer $B$*, and a set of *dependency arcs $A$*. The stack maintains partial parses, the buffer stores the words to be parsed, and $A$ keeps the arcs that have been generated so far. The configuration of the parser at each stage can be defined by a triple $(S, B, A)$. For the ease of notation, we use $w_p$ to represent the leftmost word in the buffer and use $s_1$ and $s_2$ to denote the top and the second top words in the stack. A dependency arc $(w_h, w_m)$ is a directed edge that indicates word $w_h$ is the parent of word $w_m$. When the parser terminates, the arcs in $A$ form a projective dependency tree. We assume that each word only has one parent in the derived dependency parse

---

**Algorithm 7** TRANS($S$, $B$, $A$, action)

---
1: Let $w_p$ be the leftmost element in $B$
2: **if** action = SHIFT **then**
3:     $S$.push($w_p$)
4:     remove $w_p$ from $B$
5: **else if** action= REDUCE-LEFT **then**
6:     $top \leftarrow S$.pop()
7:     $A \leftarrow A\cup (w_p,\text{top})$
8: **else if** action = REDUCE-RIGHT **then**
9:     $top \leftarrow S$.pop()
10:     $A \leftarrow A\cup (S.\text{top}(), \text{top})$
11: **return** $S, B, A$

---

tree, and use $A[w_m]$ to denote the parent of word $w_m$. For labeled dependency parsing, we further assign a tag to each arc representing the dependency type between the head and the modifier. For simplicity, we assume an unlabeled parser in the following description. The extension from an unlabeled parser to a labeled parser is straightforward, and is discussed at the end of this section.

In the following, we describe an arc-hybrid transition system due to its simplicity. The arc-eager system used in the experiments share the same spirit. In the initial configuration, the buffer $B$ contains all the words in the sentence, a dummy root node is pushed in the stack $S$, and the set of arcs $A$ is empty. The root node cannot be popped out at anytime during parsing. The system then takes a sequence of actions until the buffer is empty and the stack contains only the root node (i.e., $|B| = 0$ and $S = \{\mathbf{Root}\}$). When the process terminates, a parse tree is derived. At each state, the system can take one of the following actions:

1. SHIFT: push $w_p$ to $S$ and move $p$ to the next word. (Valid when $|B| > 0$).
2. REDUCE-LEFT: add an arc $(w_p, s_1)$ to $A$ and pop $s_1$. (Valid when $|B| > 0$ and $|S| > 1$).
3. REDUCE-RIGHT: add an arc $(s_2, s_1)$ to $A$ and pop $s_1$. (Valid when $|S| > 1$).

Algorithm 7 shows the execution of these actions during parsing, and Figure 5 demonstrates an example of transition-based dependency parsing.

## C    Additional Experiment Results

### C.1    Sequential Tagging

In Figure 6, we enlarge the figures in 3 and provide the results of NER with default parameters.

### C.2    Dependency Parsing

Table 2 show the complete experiment results for dependency parsing. The system is evaluated on both unlabeled attachment score (UAS) and labeled attachment score. Again, conducting fair comparisons across different systems is difficult because different systems use different sets of features and different assumptions about the structure of languages. Table 3 summarizes the differences.

## D    Experiment details

### D.1    Datasets and Tasks

We conduct experiments based on two variants of the sequence labeling problem (Algorithm 1) The first is a pure sequence labeling problem: Part of Speech tagging based on data from the Wall Street Journal portion of the Penn Treebank. The second is a sequence *chunking* problem: named entity

Figure 6: Training time versus evaluation accuracy for part of speech tagging (left) and named entity recognition (right). X-axis is in log scale. Different points correspond to different termination criteria for training. Top figures use hyperparameters that were tuned (for accuracy) on the holdout data. (Note: lines are curved due to log scale x-axis.)

| Parser | Ar | Bu | Ch | Cz+ | Da | Du+ | Ja+ | Po+ | Sl+ | Sw | PTB | CTB |
|---|---|---|---|---|---|---|---|---|---|---|---|---|
| | | | | | | UAS | | | | | | |
| DYNA | 75.3 | 89.8 | 88.7 | 81.5 | 87.9 | 74.2 | 92.1 | 88.9 | 78.5 | 88.9 | 90.3 | 80.0 |
| SNN | 67.4* | 88.1 | 87.3 | 78.2 | 83.0 | 75.3 | 89.5 | 83.2* | 63.6* | 85.7 | 91.8[#] | 83.9[#] |
| L2S$^O$ | 75.3 | 89.5 | 87.4 | 81.1 | 86.0 | 75.3 | 90.4 | 88.4 | 78.5 | 89.9 | 91.9 | 85.1 |
| L2S | 78.2 | 92.0 | 89.8 | 84.8 | 89.8 | 79.2 | 91.8 | 90.6 | 82.2 | 89.7 | 91.9 | 85.1 |
| Best | 79.3 | 92.0 | 93.2 | 87.30 | 90.6 | 83.6 | 93.2 | 91.4 | 83.2 | 89.5 | 94.4[#] | 87.2[#] |
| | | | | | | LAS | | | | | | |
| DYNA | 64.3 | 85.0 | 84.6 | 74.1 | 82.5 | 70.3 | 90.6 | 85.0 | 68.5 | 83.5 | 88.1 | 78.8 |
| SNN | 51.7* | 84.0 | 82.7 | 77.4 | 72.0 | 89.1 | 87.4* | 77.9* | 51.1* | 80.1 | 89.6[#] | 82.4[#] |
| L2S$^O$ | 65.1 | 85.0 | 80.8 | 74.5 | 81.0 | 72.1 | 88.4 | 84.4 | 69.4 | 85.2 | 89.7 | 83.6 |
| L2S | 68.2 | 88.2 | 87.1 | 79.6 | 84.9 | 75.8 | 89.7 | 87.8 | 74.0 | 84.9 | 89.7 | 83.6 |
| Best | 66.9 | 87.6 | 90.0 | 80.2 | 84.8 | 79.2 | 91.7 | 87.6 | 73.4* | 84.6 | 92.55[#] | 85.7[#] |

Table 2: Accuracy on PTB, CTB and CoNLL-X. Best: best results from the shared task. Best: the best results reported in CoNLL-X (may come from different participants) and the best published results (CTB,PTB). L2S$^O$, DYNA, SNN use only features generated by word and POS tags, while L2S and the models in CoNLL-X use additional morphologic features. L2S models use Brown cluster for PTB, DYNA and the Best use word embedding features generated from unsupervised text corpus with billion words. The Best models also used word embedding features for $CTB$.[19]

| Parser | Ar | Bu | Ch | Cz$^+$ | Da | Du$^+$ | Ja$^+$ | Po$^+$ | Sl$^+$ | Sw | PTB | CTB |
|---|---|---|---|---|---|---|---|---|---|---|---|---|
| Dyna |  |  | r |  | r | r | r | r |  |  | T | T |
| Snn | r' |  | r |  | r | r | r' | r' |  |  | T | T |
| L2S$^O$ |  |  | r |  | r | r | r | r |  |  | T | T |
| L2S | M | M | M | rM | M | rM | rM | rM | rM | M | ET | T |
| Best | TM | TM | TM | TM | TM | TM | TM | TM | TM | TM | ET | ET |

Table 3: T: tuned hyper-parameters, M: use morphological features, E: use word embedding or word clustering, r: language structure assumption that may degrades the performance (the nature of transition-based model), r': strong language structure assumption (only one head) that severely degrades the performance. Accuracy on PTB, CTB and CoNLL-X. Best: best results from the shared

|  | Training | | | | | Holdout | | Test | |
|---|---|---|---|---|---|---|---|---|---|
|  | Sents | Toks | Labels | Features | Unique Fts | Sents | Toks | Sents | Toks |
| **POS** | 38k | 912k | 45 | 13,685k | 629k | 5.5k | 132k | 5.5k | 130k |
| **NER** | 15k | 205k | 7 | 8,592k | 347k | 3.5k | 52k | 3.6k | 47k |

Table 4: Basic statistics about the data sets used for part of speech (POS) tagging and named entity recognition (NER).

recognition using data from the CoNLL 2003 dataset. See Figure 7 for example inputs and outputs for these tasks.

Part of speech tagging for English is based on the Penn Treebank tagset that includes 45 discrete labels. The accuracy reported represents number of tokens tagged correctly. This is a *pure* sequence labeling task. Named entity recognition for English is based on the CoNLL 2003 dataset that includes four entity types: Person, Organization, Location and Miscellaneous. We use the standard evaluation metric to report performance as macro-averaged F-measure. In order to cast this *chunking* task as a sequence labeling task, we use the standard Begin-In-Out (BIO) encoding, though some results suggest other encodings may be preferable [37] (we tried BILOU and our accuracies decreased). The example sentence from Figure 7 in this encoding is:

<div align="center">

LOC      ORG      PER

Germany 's rep to the European Union 's committee Werner Zwingmann said ...

B-LOC  O  O  O  O  B-ORG  I-ORG  O  O  B-PER  I-PER  O

</div>

Dependency parser is test on the English Penn Treebank (PTB) and the CoNLL-X datasets for 9 other languages, including Arabic, Bulgarian, Chinese, Danish, Dutch, Japanese, Portuguese, Slovene and Swedish. For PTB, we convert the constituency trees to dependencies by the Stanford parser 3.3.0. We follow the standard split: sections 2 to 21 for training, section 22 for development and section 23 for testing. The POS tags in the evaluation data is assigned by the Stanford POS tagger, which has an accuracy of 97.2% on the PTB test set. For CoNLL-X, we use the given train/test splits and reserve the last 10% of training data for development if needed. The gold POS tags given in the CoNLL-X datasets are used. The CTB is prepared following the instructions in [8].

### D.2  Methodology

Comparing different systems is challenging because one wishes to hold constant as many variables as possible. In particular, we want to control for both **features** and **hyperparameters**. In general, if a methodological decision cannot be made "fairly," we made it in favor of competing approaches.

| **POS** | NNP  NNP  , CD NNS  JJ , MD VB DT  NN  IN DT  JJ  NN<br>Pierre Vinken , 61 years old , will join the board as  a  nonexecutive director ... |
|---|---|
| **NER** | LOC    ORG    PER<br>Germany 's rep to the European Union 's committee Werner Zwingmann said ... |

Figure 7: Example inputs (below, black) and desired outputs (above, blue) for part of speech tagging task, named entity recognition task, and entity-relation recognition task.

To control for **features**, for the two sequential tagging tasks (POS and NER), we use the built-in *feature template* approach of CRF++ (duplicated in CRF SGD) to generate features. The other approaches (Structured SVM, VW Search and VW Classification) all use the features generated (offline) by CRF++. For each task, we tested six feature templates and picked the one with best development performance using CRF++. The templates included neighboring words and, in the case of NER, neighboring POS tags. *However,* because VW Search is also able to generate features from its own templates, we also provide results for **VW Search (own fts)** in which it uses its own, internal, feature template generation, which were tuned to maximize it's holdout performance on the most time-consuming run (4 passes) and include neighboring words (and POS tags, for NER) and word prefixes/suffixes.[20] In all cases we use *first order Markov dependencies,* which lessens the speed advantage of search based structured prediction.

To control for **hyperparameters**, we first separated each system's hyperparameters into two sets: (1) those that affect termination condition and (2) those that otherwise affect model performance. When available, we tune hyperparameters for (a) learning rate and (b) regularization strength[21]. Additionally, we vary the termination conditions to sweep across different amounts of time spent training. For each termination condition, we can compute results using either the **default hyperparameters** or the **tuned hyperparameters** that achieved best performance on holdout data. We report both conditions to give a sense of how sensitive each approach is to the setting of hyperparameters (the amount of hyperparameter tuning directly affects effective training time).

One final confounding issue is that of **parallelization**. Of the baseline approaches, only CRF++ supports parallelization via multiple threads at training time. In our reported results, CRF++'s time is the total CPU time (i.e., effectively using only one thread). Experimentally, we found that wall clock time could be decreased by a factor of $1.8$ by using $2$ threads, a factor of $3$ using $4$ threads, and a (plateaued) factor of $4$ using $8$ threads. This should be kept in mind when interpreting results. DEMI-DCD (for structured SVMs) also *must* use multiple threads. To be as fair as possible, we used $2$ threads. Likewise, it can be sped up more using more threads [6]. VW (Search and Classification) can also easily be parallelized using AllReduce [1]. We do not conduct experiments with this option here because none of our training times warranted parallelization (a few minutes to train, max).

For dependency parsing, we fixed the hyper-parameters when test on CoNLL-X. For CTB and PTB, we tune the size of beam in beam search and the history length of predictions. For PTB, we further use dictionary features from Brown cluster.

### D.3 Hardware Used

All timing results were obtained on the same machine with the following configuration. Nothing else was run on this machine concurrently:

```
%   2 * Intel(R) Core(TM)2 Duo CPU E8500 @ 3.16GHz
  6144 KB cache
  8 GB RAM, 4 GB Swap
  Red Hat Enterprise Linux Workstation release 6.5 (Santiago)
  Linux 2.6.32-431.17.1.el6.x86_64 #1 SMP
    from Fri Apr 11 17:27:00 EDT 2014 x86_64 x86_64 x86_64 GNU/Linux
```

### D.4 Software Used

The precise software versions used for comparison are:

CRF++  The popular CRF++ toolkit [27] for conditional random fields [28].
CRF SGD  A stochastic gradient descent conditional random field package [4].
Structured Perceptron  An implementation of the structured perceptron [9] due to [6].
- ] The cutting-plane implementation [23] of the structured SVMs [47] for "HMM" problems.
Structured SVM (DEMI-DCD)  A multicore algorithm for optimizing structured SVMs called DEcoupled Model-update and Inference with Dual Coordinate Descent.

Our approach is implemented in the Vowpal Wabbit [29] toolkit on top of a cost-sensitive classifier [3] that reduces to regression trained with an online rule incorporating AdaGrad [15], per-feature normalized updates [42], and importance invariant updates [24].

VW Classification An *unstructured* baseline that predicts each label independently, using one-against-all multiclass classification [3].

- latest Vowpal Wabbit version (May 2016) commit 2dfb1225c8b89c14552932161b95237fc90b636c
- CRF++ version 0.58
- crfsgd version 2.0
- svm_hmm_learn version 3.10, 14.08.08
  includes SVM-struct V3.10 for learning complex outputs, 14.08.08
  includes SVM-light V6.20 quadratic optimizer, 14.08.08
- Illinois-SL version 0.2.2

### D.5 Hyperparameters Tuned

The hyperparameters tuned and the values we considered for each system are:

### CRF++

```
%    termination parameters:
     number of passes (--max_iter)    { 2, 4, 8, 16, 32, 64, 128 }
     termination criteria (--eta)     0.000000000001 (to prevent termination)

  tuned hyperparameters (default is *):
     learning algorithm (--algorithm)  { CRF*, MIRA }
     cost parameter (--cost)           { 0.0625, 0.125, 0.25, 0.5, 1*, 2, 4, 8, 16 }
```

### CRF SGD

```
%    termination parameters:
     number of passes (-r)             { 1, 2, 4, 8, 16, 32, 64, 128 }

  tuned hyperparameters (default is *):
     regularization parameter (-c)     { 0.0625, 0.125, 0.25, 0.5, 1*, 2, 4, 8, 16 }
     learning rate (-s)                { auto*, 0.1, 0.2, 0.5, 1, 2, 5 }
```

### Structured SVM

```
%    termination parameters:
     epsilon tolerance (-e)            { 4, 2, 1, 0.5, 0.1, 0.05, 0.01, 0.005, 0.001 }

  tuned hyperparameters (default is *):
     regularization parameter (-c)     { 0.0625, 0.125, 0.25, 0.5, 1*, 2, 4, 8, 16 }
```

### Structured Perceptron

```
%    termination parameters:
     number of passes (MAX_NUM_ITER)   { 1, 2, 4, 8, 16, 32, 64, 128 }

  tuned hyperparameters (default is *):
     NONE
```

### Structured SVM (DEMI-DCD)

```
%    termination parameters:
     number of passes (MAX_NUM_ITER)   { 1, 2, 4, 8, 16, 32, 64, 128 }

  tuned hyperparameters (default is *):
     regularization (C_FOR_STRUCTURE)  { 0.01, 0.05, 0.1*, 0.5, 1.0 }
```

### L2S

```
  termination parameters:
     number of passes (--passes)       { 0.01, 0.02, 0.05, 0.1, 0.2, 0.5, 1, 2, 4 }
     (note: a number of passes < 1 means that we perform one full pass, but
      _subsample_ the training positions for each sequence at the given rate)

  tuned hyperparameters (default is *):
     base classifier                   { csoaa*}
     interpolation rate                10^{-10, -9, -8, -7, -6 }
```

**VW Classifier**

```
termination parameters:
  number of passes (--passes)      { 1, 2, 4 }

tuned hyperparameters (default is *):
  learning rate (-l)               { 0.25, 0.5*, 1.0 }
```

# E  Templates Used

For part of speech tagging (CRF++):

```
U00:%x[-2,0]
U01:%x[-1,0]
U02:%x[0,0]
U03:%x[1,0]
U04:%x[2,0]
```

For named entity recognition (CRF++):

```
U00:%x[-2,0]
U01:%x[-1,0]
U02:%x[0,0]
U03:%x[1,0]
U04:%x[2,0]

U10:%x[-2,1]
U11:%x[-1,1]
U12:%x[0,1]
U13:%x[1,1]
U14:%x[2,1]

U15:%x[-2,1]/%x[-1,1]
U16:%x[-1,1]/%x[0,1]
U17:%x[0,1]/%x[1,1]
U18:%x[1,1]/%x[2,1]
```

(where words are in position 0 and POS is in 1)

Additional features for L2S (ft) on POS Tagging:

```
-- the left and the right tokens of each word
-- the first and the last 2 characters for each token
```

For L2S (ft) on NER:

```
-- the left and the right two tokens of each word
-- the POS tags of the left and the right tokens for each word
-- the last charaster for each token
```

## Footnotes

[19](*) SNN makes assumptions about the structure of languages and hence obtains substantially worse performance on languages with multi-root trees. (+) Languages contains more than 1% non-projective arcs, where a transition-based parser (e.g. L2S) likely underperforms graph-based parser (Best) due to the model assumptions. (#) Numbers reported in the published papers [8, 16, 2].

[20]The exact templates used are provided in the supplementary materials.

[21]Precise details of hyperparameters tuned and their ranges is in the supplementary materials.