[Reviews · NeurIPS 2016]

Reviewer 1

Summary

This paper presents a system designed to speed up the implementation and execution of learning-to-search algorithms. Once the user implements their application as a small program built atop two library functions ("Predict" and "Loss"), the system can use that program to both learn the model parameters and generate predictions at test time, essentially by swapping in different versions of the two library primitives. Experiments show that several NLP problems can be solved very quickly and accurately using this system, which the authors call a "credit assignment compiler".

Qualitative Assessment

This is an interesting and novel paper. It's sort of a fusion between proposing a general "master algorithm" for a number of existing learn-to-search approaches (Searn, DAgger, etc.) and an engineering exercise in making this master algorithm run quickly. The empirical runtime/accuracy results are compelling, and I suspect the system might be useful to a wide range of people once it is released. At the same time, I have some reservations about the way the paper is written and structured. Since the authors are emphasizing that their implementation of this general algorithm is easy and fast for practitioners to use, it seems problematic that it is (to me) very difficult to understand how one would actually use it. Partly this is because, for instance, Algorithm 1 (MyRun) uses a bunch of terms that have not been explained at that point (tags, etc.) -- i.e., some reordering might help. But partly it's because some things are never really explained. For instance, other than reverse-engineering the desired learning procedure using Algorithm 3, how should a user write MyRun? How should we think of Predict and Loss in an intuitive way? And how is this different from writing a search algorithm? The concept of a TDOLR function is defined without much context, but seems to be central. There is a footnote about letting MyRun control the machine stack, but I am not sure what this is supposed to mean for pseudocode. And to be honest, I still don't understand why this is called a credit assignment compiler. I get that search provides credit assignment by measuring the effect of each decision in (relative) isolation, but in what sense is this a compiler? In general, I found the idea of the paper and the results produced compelling, but wished the middle part was better able to articulate why I should believe that this system is natural or easy to use. Some additional examples, tutorial-style, would go a long way. There are also a large number of typos and grammatical errors that need to be fixed.

Confidence in this Review

2-Confident (read it all; understood it all reasonably well)


Reviewer 2

Summary

Existing learning to search approaches are often difficult to use because defining an appropriate state space is challenging. This paper presents a new abstraction that allows learning to search problems to be easily specified with two functions: PREDICT(...) and LOSS(...). Two optimization improvements are incorporated into the library that supports this abstraction: memoization and path collapse, which reduce recomputation of predictions and the effective search space explored. An extensive set of experiments conducted on NER/POS tasks shows the benefit of the approach compared to existing structured prediction methods (CRF/SSVM) and competitive performance with specialized state-of-the-art methods for these tasks.

Qualitative Assessment

The main strength of the paper is the experimental evaluation, which shows that the approach is highly efficient while providing comparable (or better) performance with slower structured prediction methods. The compactness of the implementations (with library support) for these tasks are also impressive and could lead to wide adoption in the community, though the accessibility of the library to others remains to be seen. A broader discussion of relations with existing learning to search approaches is warranted. “Some fit this framework.” Which? What existing capabilities in these approaches are difficult to incorporate into the framework? The experiments were run on outdated hardware (Core 2 Duo E8500) that is ~8 years old with only 8GB of memory. Comparisons using modern CPU architectures that support much more parallelization are needed if the point is to encourage adoption.

Confidence in this Review

2-Confident (read it all; understood it all reasonably well)


Reviewer 3

Summary

A good NIPS paper, assuming the authors release the implementation. The authors present L2S, which seems like a toolbox to implement all previous learning to search style algorithms. It has two optimizations to make it easy for users to speed up training when the model allows it, and provides decent results on several tasks.

Qualitative Assessment

Strengths: + Clear presentation. Easy to read and understand. + Clear idea: very simple algorithm, with clear exposition of the two optimizations. + Good enough results: POS tagging, Dependency parsing aren't state of the art, but they're comparable to similar approaches. They are good enough that this framework would be useful to have for the NIPS community. Weakness: (e.g., why I am recommending poster, and not oral) - Impact: This paper makes it easier to train models using learning to search, but it doesn't really advance state-of-the-art in terms of the kind of models we can build. - Impact: This paper could be improved by explicitly showing the settings for the various knobs of this algorithm to mimic prior work: Dagger, searn, etc...it would help the community by providing a single review of the various advances in this area. - (Minor issue) What's up with Figure 3? "OAA" is never referenced in the body text. It looks like there's more content in the appendix that is missing here, or the caption is out of date.

Confidence in this Review

2-Confident (read it all; understood it all reasonably well)


Reviewer 4

Summary

This paper considers the development of a compiler for classifier-based structured prediction algorithms. The goal of compiler is to take the high-level implementation of greedy classifier-based inference procedure and automatically produce the corresponding training code for advanced imitation learning algorithms (e.g., SEARN, DAgger, AggreVaTe, and LOLS).

Qualitative Assessment

Summary: This paper considers the development of a compiler for classifier-based structured prediction algorithms. The goal of compiler is to take the high-level implementation of greedy classifier-based inference procedure and automatically produce the corresponding training code for advanced imitation learning algorithms (e.g., SEARN, DAgger, AggreVaTe, and LOLS). Pros: - Very nice research direction with potential for significant impact. - Good experimental results on sequence labeling and dependency parsing tasks. Cons: - Low on the technical novelty. - Exposition can be improved in some parts of the paper. Detailed Comments: - I really like the problem considered in the paper, but unfortunately, the paper is low on the technical novelty aspect. - My understanding is that compiler will also need the specification of an oracle or reference policy in addition to "MyRun" (inference algorithm), correct? In fact, this reference policy can be complex for non-decomposable loss functions (modulo the theory in LOLS paper). For example, to the best of my knowledge, the progress on dependency parsing using similar ideas was made only after Yoav Goldberg came up with "dynamic oracle" for dependency parsing. - The two optimizations to minimize the computation makes sense. It may be beneficial to consider active learning ideas to be more selective about the states for which additional cost-sensitive examples will be generated. This direction may be able to mitigate the novelty concern.

Confidence in this Review

3-Expert (read the paper in detail, know the area, quite certain of my opinion)


Reviewer 5

Summary

This paper presents a novel framework for structured prediction (which they call "joint prediction") using principles which appear to be from reinforcement learning. Its main motivation seems to be having a learning procedure which is general as possible, one which can be rapidly adapted for new tasks with very little new code. The authors were successful in this regard, performing comparably to state-of-the-art task-specific training methods on several structured prediction tasks.

Qualitative Assessment

This paper presents a novel framework for structured prediction (which they call "joint prediction") using principles which appear to be from reinforcement learning. Its main motivation seems to be having a learning procedure which is general as possible, one which can be rapidly adapted for new tasks with very little new code. The authors were successful in this regard, performing comparably to state-of-the-art task-specific training methods on several structured prediction tasks. The learning procedure consists essentially in iteratively tweaking the current-model policy by successively varying the action for a given state, and seeing if the overall loss could be improved by changing the action. This is a sensible and easy to understand approach, but the learning algorithm itself seems to be almost an afterthought in the paper. I would like to see like to see a bit more discussion of this procedure and the intuitions behind it: why was it chosen and how does it relate to the idea of "credit assignment." Overall, I think this paper is quite good. The notion of having a general framework for structured prediction which can easily be adapted to new tasks is a very good one, and likely to be useful as people continue to apply machine learning in more and more domains. The experiments are convincing insofar as they show that this approach can perform on par with learning algorithms which were developed for specific tasks.

Confidence in this Review

2-Confident (read it all; understood it all reasonably well)


Reviewer 6

Summary

The authors show that the problem of learning to search (L2S) can be turned into a credit assignment compiler. This is possible because the connection between the search spaces and terminal discrete oracle loss reporting programs. Optimizing the credit assignment complier instead leads to reduced execution and programming time.

Qualitative Assessment

The idea is interesting, but I think the writing can be improved to convey the idea better to graphical model audience. Another way to reduce execution time (both training and testing) is to use classifier chains/trees, which sometimes obtain near graphical model accuracy with fraction of the run-time. How does the proposed method compare to them? ==Post rebuttal == After seeing the authors' response and having discussions with other reviewers, my original rating remains unchanged.

Confidence in this Review

1-Less confident (might not have understood significant parts)